# Animal Models for Understanding the Mechanisms of Beta Cell Death during Type 2 Diabetes Pathogenesis

**DOI:** 10.3390/biomedicines12030473

**Published:** 2024-02-20

**Authors:** Brittney A. Covington, Wenbiao Chen

**Affiliations:** Department of Molecular Physiology & Biophysics, Vanderbilt University, Nashville, TN 37232, USA; brittney.a.covington@vanderbilt.edu

**Keywords:** type 2 diabetes, beta cell death, islet inflammation, animal models, zebrafish

## Abstract

Type 2 diabetes (T2D) has become a worldwide epidemic, primarily driven by obesity from overnutrition and sedentariness. Recent results reveal there is heterogeneity in both pathology and treatment responses in T2D patients. Therefore, a variety of T2D animal models are necessary to obtain a mechanistic understanding of distinct disease processes. T2D results from insufficient insulin, either due to beta cell loss or inborn deficiency. Although decreases in beta cell mass can occur through loss of identity or cell death, in this review, we will highlight the T2D animal models that display beta cell death, including the Zucker Diabetic Fatty Rat, sand rat, db/db mouse, and a novel diabetic zebrafish model, the Zebrafish Muscle Insulin-Resistant (zMIR) fish. Procuring a mechanistic understanding of different T2D progression trajectories under a variety of contexts is paramount for developing and testing more individualized treatments.

## 1. Introduction

Hyperinsulinemia and insulin resistance are major risk factors for the leading causes of death in the world, including cardiovascular disease and cancer [1]. These pathological phenotypes are often prerequisites for the development of type 2 diabetes (T2D). Physiologically, T2D manifests as an inability of the pancreatic beta cells to produce and secrete a sufficient bolus of insulin to elicit a response in target cells to transport glucose from the blood and properly regulate glucose levels. This leads to an elevated fasting glucose measurement of 125 mg/dL or higher in humans [2]. T2D is a worldwide epidemic, exerting substantial disruptions on economic productivity, financial stability, longevity, and health span within the afflicted population. According to the CDC’s 2019 estimates, 14.7% of the adult populace is diabetic, with another 38% being pre-diabetic. T2D is not solely an adult disease. In the past two decades, the prevalence of pediatric T2D in the United States has doubled [3]. Despite these staggering figures, if these trends persist, pediatric T2D incidences are projected to rise 4-fold by 2050, and worldwide diabetes prevalence in adults to double, with a projected 1.31 billion individuals living with diabetes by 2050 [4].

Diabetes is associated with a reduction in life span by 6 years in individuals who have diabetes at 50 years old compared to non-diabetic adults [5]. Just as concerning as the loss of lifespan, diabetes also decreases health span. In this context, health span refers to the number of years lived without significant disease and disability. Key T2D comorbidities include hypertension, ischemic heart disease, kidney disease, cancer, asthma, back pain, and osteoarthritis. As many as 37% of T2D persons at diagnosis have hypertension [6]. Additionally, diabetes is the leading cause of non-traumatic lower limb amputation, kidney failure, and new incidences of blindness in adults.

Along with adverse physical outcomes for individuals with T2D, there are detrimental psychological consequences linked with diabetes. Anxiety and depression are associated with diabetes [7]. The risk of developing depression is 50–100% higher in persons with diabetes than in the general public [8,9]. Similarly, significant correlations have been identified between psychotic disorders, including schizophrenia, and diabetes [7]. Likely due to the negative psychological and physical repercussions experienced by individuals with diabetes, there is a higher propensity for suicide in this population, with the risk nearly doubling in comparison to their non-diabetic counterparts [10,11].

Diabetes is a complex and heterogeneous medical condition characterized by a chronic elevation in blood glucose levels. In the case of T2D, the dysregulation of glucose primarily stems from two factors: (1) inadequate production of insulin by the pancreatic beta cells and/or (2) a reduction in insulin’s effectiveness to stimulate target cells to take up glucose, a condition known as insulin resistance [12]. Insulin resistance is a major hallmark of T2D and occurs when the target cells of insulin, including adipocytes, hepatocytes, and myocytes, become less sensitive and require a higher concentration of insulin to stimulate the cell to uptake glucose from the blood into the cell for storage and oxidation. Insulin resistance can occur through a plethora of mechanisms. However, obesity is a major contributor to insulin resistance and hyperinsulinemia as it instigates inflammation and consequently increases circulating FFA, adipocytes, and proinflammatory cytokines that impair insulin signaling [13]. Hyperinsulinemia itself can also contribute to insulin resistance via a common process known as homologous desensitization, whereby continuously high levels of a ligand can inhibit the responsiveness of its receptor [14]. Thereby, obesity and insulin resistance can build upon the pathophysiology of one another to culminate in T2D.

Insulin is synthesized in the endoplasmic reticulum (ER) of pancreatic beta cells where it undergoes a series of post-translational modifications to form mature insulin. Insulin resistance requires more insulin to be produced by beta cells to compensate for these desensitized cells. Consequently, this compensation causes additional strain on beta cells [15,16]. This stress primarily originates from the ER and can also trigger oxidative stress [17,18]. These cellular stresses can lead to beta cell decompensation, manifested by dysfunction and eventually a loss of beta cell mass.

Unfortunately, though T2D presents as an inability to regulate glucose levels in the appropriate range, there are a multitude of factors that can influence an individual’s propensity to develop T2D, with the process being much more complicated than solely calories in verses calories out. While an extensive body of scientific literature exists within the domain of diabetes, a considerable realm of undiscovered knowledge still remains, vastly eclipsing what has already been uncovered. Given the substantial impact of diabetes on financial well being, reductions in both life and health span, adverse psychological effects, and an alarmingly elevated incidence of suicide among diabetic patients, it becomes imperative to focus on enhancing our comprehension of the complex pathophysiology of diabetes. As mechanisms regulating glucose homeostasis are evolutionarily conserved, exploring disease pathogenesis across different diabetic vertebrate models may uncover novel pathways of disease development, providing a better understanding of this multifaceted disease and potentially paving the way for more personalized treatments for patients. Indeed, animal models have already been instrumental in the development of most pharmaceutical drugs, including the Nobel prize-winning discovery of insulin utilizing a dog model. This review will explore diabetic hallmarks using animal models in four different species, highlighting a novel zebrafish diabetic model.

### Current Drug Treatments

Although metformin and thiazolidinediones are still commonly used, several new drug classes have been approved to treat T2D over the past two decades. These include non-sulfonylurea K_ATP_ antagonists, alpha-glucosidase inhibitors, DPP-4 inhibitors, GLP-1 receptor agonists (GLP-1RAs), and SGLT2 inhibitors [19]. These diabetic drugs exhibit distinct mechanisms of action in managing blood glucose levels [19,20]. Metformin and thiazolidinediones (such as pioglitazone and rosiglitazone) are insulin sensitizers that improve insulin sensitivity in peripheral tissues, such as liver, muscle, and adipose (fat) tissue [21,22]. Non-sulfonylurea K_ATP_ antagonists, including the medications nateglinide and repaglinide, stimulate insulin release by targeting ATP-sensitive potassium channels on pancreatic beta cells, promoting calcium influx and insulin secretion [23,24]. Alpha-glucosidase inhibitors, such as acarbose, exert their effects by slowing down carbohydrate digestion in the small intestine, leading to a gradual release of glucose after meals [25,26]. SGLT2 inhibitors, including canagliflozin, reduce renal glucose reabsorption, increasing urinary glucose excretion and providing potential cardiovascular and renal benefits [27].

DPP-4 inhibitors and GLP-1RAs both act on GLP-1R, the receptor for GLP-1. GLP-1 is a hormone secreted from enteroendocrine L-cells in response to food [28]. GLP-1 has been shown to increase insulin sensitivity by elevating the expression of GLUT 4, the glucose transporter, in insulin-dependent tissues [29,30]. GLP-1 acts at the level of the pancreas to promote beta cell proliferation and insulin release while inhibiting glucagon secretion [28,30,31]. GLP-1 can also improve insulin sensitivity by promoting weight loss [32,33]. This is mainly through GLP-1′s effect in the hypothalamus, whereby GLP-1 reduces feelings of hunger, thereby decreasing food intake [32]. GLP-1 also reduces food intake by slowing gastric emptying, but the mechanism remains unclear [30,31]. DPP-4 inhibitors, including sitagliptin, saxagliptin, linagliptin, and alogliptin, increase GLP-1 levels by inhibiting its degradation [34]. GLP-1RAs, including semaglutide, exenatide, and liraglutide, are long-acting GLP-1 analogs that activate GLP-1Rs supraphysiologically [35]. GLP-1RAs have recently surged in popularity due to their potent weight loss effect. As a result, they also ameliorate insulin resistance in T2D patients [30].

Despite the multitude of drug targets and treatment options, clinical trials consistently find approximately half of the enrolled patients fail to reach the ADA-recommended goal of an HbA1c ≤ 7% [36,37,38,39,40]. Furthermore, all current treatments have side effects [41,42,43]. Insulin and sulfonylureas are known to cause weight gain and hypoglycemia. On the other hand, thiazolidinediones are linked to weight gain, edema, and an elevated risk of cardiac events and bone fractures in women. Some thiazolidinediones have been withdrawn from use due to these adverse effects. Though GLP-1 RA drugs, including liraglutide and semaglutide, have been shown to be effective in reducing weight and decreasing insulin resistance and hyperglycemia, it is important to note that some individuals may experience gastrointestinal issues or develop other adverse health outcomes as an off-target effect [31,44,45].

## 2. A Heterogeneous Disease

A multitude of factors contribute to T2D development in humans. Some of these elements include ethnicity, exposure to toxins, circadian disruptions, stress, activity level, epigenetic, genetic, and even microbiota variations as shown in Figure 1 [9,46,47,48,49,50]. As such, T2D is heterogenous in etiology and phenotype. Here, we briefly review recent evidence on genetic and phenotypic heterogeneity.

T2D has been classified into different subgroups based on phenotypes. Several classification schemes have been reported [51,52,53,54,55]. For example, Ahlqvist et al. assigned T2D patients into four clusters, including severe insulin-deficient diabetes (SIDD), severe insulin-resistant diabetes (SIRD), mild obesity-related diabetes (MOD), and mild age-related diabetes (MARD) [51]. More recently, Nair et al. projected nine T2D-related phenotypes at diagnosis of 23,137 Scottish T2D patients into a tree with seven branches through a dimension reduction DDRTree algorithm. Each branch is distinct in the strength of the nine phenotypes [56].

GWAS studies have identified more than 400 genes that are associated with T2D [57]. These variants only explain approximately 20% of the diabetes risk, less than half of the estimated heritability of T2D in the European population [57,58]. Therefore, more T2D-associated genes remain to be identified. By examining the correlation of the genetic architecture of T2D with other related traits, Mahajan et al. identified a link between T2D risk and sleeping behaviors, smoking, metabolites, depressive symptoms, urinary albumin-to-creatinine ratio, and urate [57]. By assigning risk alleles into groups of likely pathogenic pathways, Udler et al. identified five different clusters [59]. Two clusters have indications of reduced beta cell function. The other three clusters displayed features of insulin resistance, “lipodystrophy-like” fat distribution, and disrupted liver lipid metabolism.

The T2D classifications also have implications for complication development and treatment options. As an example, the SIRD cluster had a substantially higher risk of developing diabetic kidney disease than the other clusters [51]. Similarly, patients in different branches in the Nair et al. study respond differently to the wide variety of drug treatments available and have divergent propensities to develop various complications [60]. Nonetheless, simple lab tests outperformed the cluster assignment in selecting drug and treatment plans for patients [26,56,60].

Overall, these studies show the importance of treating diabetes as a continuum with an abundance of variation between individual patients. Deducing the most optimal treatment for diabetic patients is essential, as managing blood glucose in a euglycemic range greatly reduces the risk for adverse events, including a 40% decrease in the risk of eye, kidney, and nerve disease [61]. Therefore, it is important for researchers and clinicians to continue finding and producing optimal drugs and therapies for individual patients. Employing a diverse array of animal models that mimic variations in disease progression will enhance comprehension of pathogenic diabetic pathways, thereby fostering improved treatment strategies for patients (Figure 1).

## 3. T2D Animal Models

Animal models can be leveraged to understand diabetic mechanisms in a way that is impossible in humans today. The most common diabetic animal models are rodents. However, many different animal models have been used in diabetic research, including zebrafish, non-human primates, dogs, pigs, and sheep. Humans and other animals share a huge degree of similarity in physiology and disease processes. Interestingly, 90% of medications for animals also work in the same way in humans. Additionally, animal models are useful for determining the mechanisms of disease in a very specific context. Because humans have a vast degree of heterogeneity between individuals genetically and environmentally, it is difficult to pinpoint the mechanisms driving disease. However, animal models reduce the amount of genetic variability by conducting experiments in inbred animals and by utilizing siblings. Therefore, the researcher can manipulate one aspect at a time and determine what effects are produced. This scientific process utilizing animal models is incredibly powerful, and researchers would be decades behind if we could not use animal models. For reviews that explore a comprehensive array of animal models in diabetes, Kottaisamy et al. and Pandey et al. are excellent resources [62,63]. The following text will summarize three different mammalian diabetic models and one novel zebrafish diabetic model that all present with beta cell compensation, islet inflammation, and beta cell loss; however, there are distinct mechanisms of disease within these realms (Table 1).

### 3.1. Rodent Models

Rodent animal models are commonly used in T2D research. Islet composition and function are well conserved between rodents and humans. Rodents, particularly mice, are easy to manipulate with genetic, pharmacologic, or dietary approaches to create a desired diabetic phenotype. There are numerous models that can mimic aspects of human pathologies. For this review, we will briefly discuss selected rodent models that show beta cell compensation followed by a loss of beta cell mass, with the caveat that dozens of rodent models have different paths of disease progression.

Zucker Diabetic Fatty Rats (ZDFs): ZDFs represent a non-insulin-dependent diabetes model due to a genetic mutation in the leptin receptor [66]. ZDF rats have a missense mutation at nucleotide position 806, resulting in an amino acid switch from Gln to Pro at position 269, a region in the extracellular domain of the leptin receptor [68]. This mutation results in the inhibition of leptin signaling, leading to hyperphagic and obese rats. Hyperinsulinemia is detected at 3 weeks of age and hyperglycemia by 7 weeks for male rats [69]. At 19 weeks, the insulin amount significantly drops as the islets atrophy [69]. This model is advantageous for studying T2D progression from obesity and insulin resistance-induced beta cell compensation to decompensation, caused at least in part by beta cell death.

*Psammomys obesis* (sand rat): The sand rat is a terrestrial mammal from the gerbil subfamily that only eats the stems and leaves of plants from the Amaranth family in their natural habitat. In captivity, when fed with a high-energy diet, they develop T2D [70]. Therefore, unlike the other models in this review, the sand rat represents a non-genetic T2D model with disease being induced by diet. These animals experience insulin resistance, hyperglycemia, and a significant reduction in beta cell mass within 4–6 weeks on a calorie-dense diet. This model is advantageous in studying the role of beta cell dysfunction in T2D development, as a loss of beta cell mass occurs quickly [65]. Jörns et al. described beta cell loss as a consequence of increased necrotic beta cell death and reductions in proliferation [70].

Db/db mouse: This mouse strain represents a genetically induced T2D phenotype mouse model. This model has a 106-nucleotide insertion in the leptin receptor gene causing premature termination of the intracellular region of the leptin receptor, leading to an inhibition of leptin signaling in the hypothalamus and unchecked hyperphagia, obesity, hyperinsulinemia, and increased leptin levels [67,71,72]. Db/db mice have a significant increase in beta cell proliferation in juvenile animals, which is followed by a gradual decrease in beta cell mass later in life [73]. This model has been used to study diabetic dyslipidemia, a major factor leading to atherosclerosis [74]. Additionally, investigation of neurobehavior complications associated with T2D, including anxiety and depression, has been used in this model [75]. Because of the similarity between mice and humans, novel pharmaceutical drugs can be tested first in mice to determine their effects on feeding behaviors and other diabetic phenotypes.

### 3.2. Zebrafish Model

Zebrafish serve as excellent model organisms to study the pathophysiology of diabetes [76]. Zebrafish and humans share the same basic islet architecture, possessing identical endocrine cell types whose development involves conserved gene networks and pathways [77]. The nutrient–secretion coupling machinery for insulin secretion is conserved in zebrafish [77]. Zebrafish can develop a diabetic phenotype when exposed to prolonged overnutrition feeding [78]. Zebrafish are translucent in the early stages of development [47]. The transparent quality of zebrafish allows for live imaging of beta cells. This is a huge advantage over using a mouse model, as disease processes and changes can be seen in real time in a living organism. Zebrafish are easily genetically manipulated and produce a bolus of offspring upon breeding in a short time span, making these organisms an outstanding model for chemical and genetic screens. Zebrafish can develop enlarged fat stores, hyperglycemia, and hyperlipidemia when continuously fed a high-fat or calorie-dense diet.

Zebrafish muscle insulin resistance model (zMIR): We have established a novel insulin-resistant zebrafish model (zMIR) that possesses a dominant-negative form of the IGF-1 receptor inhibiting both insulin and IGF signaling in fast twitch muscles [64]. Upon 3 days of high-fat diet fed 5% egg yolk emulsion, larval zebrafish experience beta cell compensation and subsequent decompensation along with islet inflammation and immune cell infiltration, phenotypes also associated with T2D human patients.

### 3.3. Limitations of Animal Models

Though animal models have been essential in research progression in the diabetes field, there are many limitations that must be addressed when trying to translate discoveries in animal models into humans. Notably, humans have variations in islet structure depending on the islet size. Smaller islets have a structure similar to rodents and zebrafish, with a core of beta cells surrounded by a mantle of non-beta cells [79]. However, larger islets have a more dispersed arrangement of beta cells and a variation in the percentage of beta cells within the islet [79]. This heterogeneity may be important in disease progression and discrepancies between humans versus zebrafish and rodents. There are also variations in metabolic enzymes in humans and rodents [80]. For instance, the enzyme pyruvate carboxylase, an important protein in insulin secretion, may be up to 90% lower in human islets than in rat and mouse islets [80]. Regarding the zebrafish specifically, there is no white adipose tissue (WAT) at this early stage of development in which the zMIR diabetes-prone model is tested [64]. Additionally, because zebrafish are ectoderms, they do not possess brown adipose tissue (BAT). WAT and BAT release important cytokines that have been implicated in T2D [81,82]. Due to the small size of larval and even adult zebrafish, there are some experimental limitations that exist. Some of these include measuring insulin secretion and having the ability to perform single-cell RNA sequencing on the limited number of beta cells available.

## 4. Compensation: Beta Cell Proliferation, Transdifferentiation, and Neogenesis

The expansion of beta cell mass is infrequent in adult humans, with an estimated rate of 0.1–0.5% proliferating cells versus a 4% peak in fetal development [83,84,85]. However, under conditions of stress and heightened insulin demand due to factors including excess calorie intake, insulin resistance, and pregnancy, not only beta cell function but also beta cell mass increases to compensate for an amplified insulin need [86,87]. Beta cell mass expansion can occur through a variety of mechanisms, including neogenesis, proliferation, and transdifferentiation (Figure 2). Neogenesis arises when endocrine progenitor cells become beta cells. Proliferation refers to the replication of existing beta cells. Transdifferentiation is the shift of a non-beta cell in the islet into a beta cell. These new beta cells are functional and increase insulin content to help combat hyperglycemia and T2D [88,89]. In combination with additive factors, reductive processes, including cell death and dedifferentiation, or a loss of beta cell identity can be inhibited to promote beta cell mass expansion (Figure 2).

Beta cell compensation has been reported in various animal models, including zebrafish, rodents, non-human primates, and humans [86,87,90]. Butler et al. showed evidence of beta cell compensation in humans by their work with human cadavers. The study revealed that individuals who were classified as obese had a larger beta cell volume than individuals with (a) both obesity and T2D and (b) normal-weight individuals [90]. The results imply that obesity can induce a rise in beta cell mass, which is decreased in diabetic obese patients.

Though these results suggest a change in beta cell mass at different stages of T2D development, there are natural variations in beta cell mass in healthy individuals [91]. Furthermore, obesity state and beta cell mass do not track perfectly in all ethnicities. In a study by Inaishi and colleagues, Japanese individuals had no significant difference in beta cell mass between lean and obese subjects regardless of T2D [48]. Therefore, although there is evidence that supports beta cell mass compensation in humans until beta cell mass can be tracked from birth to disease state, there is not a definitive answer for the extent to which beta cell mass compensation occurs in humans.

### 4.1. Beta Cell Expansion in Rodent Models

Expansions in beta cell mass can occur through multiple biological pathways, including hypertrophy, transdifferentiation, neogenesis, and beta cell replication. In the diabetes field, there is a long history of debate regarding the primary pathway responsible for beta cell mass expansion in adulthood. Early studies by Bonner-Weir and colleagues found when rats were exposed to short bouts of hyperglycemia through glucose infusions, beta cell mass increased by 50% and the mitotic index, a measurement attained from the accumulation of mitotic frequency, increased by 5-fold [92]. The increase in mitotic frequency suggests that the major pathway of beta cell mass expansion was replication in this study [92].

In other rodent models, including mice, beta cell replication also appeared to be the primary means of mass expansion instead of neogenesis or endocrine transdifferentiation [93,94]. Dor et al. found via genetic lineage tracing that pre-existing beta cells rather than pluripotent stem cells were the major avenue of mass expansion in mice [94]. Indeed, Dalboge et al. and colleagues found that the total beta cell mass more than doubled in db/db mice from 5 weeks to 12 weeks of age [73]. The major route of cell expansion in these studies was concluded to stem from beta cell proliferation driven via increases in islet size and not islet number [73].

Pick et al. and colleagues found while investigating ZDF rats that at 5–7 weeks old, beta cell mass was significantly increased in the ZDF rat compared with Zucker lean control (ZLC) rats [95]. Furthermore, the increase in mass was noted as coming from proliferation, as an immunochemistry method, 6-h 5-bromo-2′-deoxyuridine (BrdU) incorporation, indicated that cell proliferation was the major source of mass increase [95].

Sand rats, when fed a high-energy diet, show dramatic increases in beta cell mass even after short 2- and 5-day overfeeding diets [96]. Interestingly, in these short-term feeding models, increases in beta cell mass mostly stemmed from increased rates of beta cell proliferation, as shown by PCNA staining. However, in sand rats fed a high-energy diet for 22 days, beta cell neogenesis increased by sixfold [96]. These studies by Kaiser et al. illustrated that the means of increased mass in sand rats may change depending on the timeline of the disease.

Although there is a bolus of studies supporting replication as a major compensatory pathway in murine animal models, other studies contest these early findings showing both neogenesis and replication in mice under different settings, including pancreatic regeneration, partial pancreatectomy, and partial duct ligation along with various drug treatments [97,98,99]. Furthermore, using control and diabetic mouse models, a new study with improved lineage tracing by Gribben and colleagues illustrated that progenitor ductal cells expressing Ngn3 contribute to adult beta cell mass in adulthood [100]. Therefore, there is still a debate on the origin of new beta cells generated during compensation. However, both neogenesis and replication probably contribute to the expansion of beta cell mass, and which is the major pathway likely depends on the individual and the specific context of the disease.

### 4.2. Human Beta Cell Expansion

Research in beta cell compensation in humans is a much harder feat than in animal models, as the major methods for assessing beta cell mass are from autopsies and organ donors. However, there are still a few studies that were able to evaluate the question of beta cell mass in humans. In a study by Butler and colleagues examining autopsies of pregnant women, relative beta cell volume was increased by 40% in pregnant versus non-pregnant women [101]. However, only a small increase in proliferation was observed in the pancreas assessed by the Ki67 marker for replication [101]. Instead of enlarged islets, which would be more indicative of replication, there was an increased number of small islets distributed around the pancreas, and insulin-positive cells were found within the ducts; therefore, neogenesis appeared to be the major pathway of expansion rather than replication [101]. Indeed, the presence of insulin-positive duct cells has been found in a few autopsied adult studies. Furthermore, in an obese model, when human islets were transplanted in mice, although there was a robust amount of native beta cell proliferation in response to a high-fat diet, there was little to no proliferation in the human islets [102]. These results suggest that beta cell expansion may follow neogenesis instead of the replication of existing beta cells. However, more work needs to be performed to definitively determine the major pathway of beta cell expansion in humans during pregnancy and in obesogenic settings.

### 4.3. Zebrafish Beta Cell Expansion

The major form of compensatory beta cell expansion in zebrafish at the larval stage may be neogenesis. Our group found that when zebrafish were treated with glucose for 8 h, beta cell mass increased by 30% in larval fish, and the increase in mass was due to the neogenesis of beta cells arising from endocrine precursor cells expressing *mnx1* or *nkx2.2* [78]. This expansion of beta cell mass is attributed to persistent insulin secretion, as prolonged pharmacologic activation of beta cell insulin secretion is sufficient to induce a compensatory response in zebrafish without feeding [103]. Interestingly, a similar mechanism also regulates beta cell proliferation in mice [104]. In contrast, drugs that block insulin secretion inhibit overnutrition-induced beta cell mass expansion [78].

Our lab also developed an insulin resistant zebrafish model (zMIR) [64]. When challenged with a high lipid diet, these fish experience beta cell compensation rapidly [64]. Beta cell mass expansion was investigated using transgenic cell markers and lineage tracing. Cell expansion was attributed to neogenesis instead of the replication of existing beta cells in the first 4 weeks of age [64]. However, at the juvenile stage, zebrafish exhibit a burst of beta cell proliferation. The proliferation is dependent on feeding [105].

Unlike humans, zebrafish have a tremendous regenerative capacity. Several groups have found that when beta cells are abolished using chemical or genetic ablation, beta cells regenerate, and the regenerated cells are functionally competent to regulate glucose levels within 1 month of insult in both larval and adult fish [106,107,108]. The origin of the regenerated cells has been widely studied and is reviewed by Yang et al. [86]. Early studies implicate alpha cells and notch-responsive ductal cells as a resource of new beta cells [109,110]. Recently, using single-cell transcriptomics and lineage tracing, the Ninov and Manfroid groups demonstrated that a major source of new beta cells after ablation is *sst1*-expressing cells in the islet [111,112]. These cells become Sst1+ Ins+ bihormonal cells and eventually Ins+ monohormonal cells [111,112,113]. Another source of new beta cells is ghrelin-expressing epsilon cells [114]. To determine the relative contribution of these sources to beta cell regeneration, Mi et al. performed a series of lineage tracing studies [113]. They found the major source of regenerated beta cells was not from alpha, delta (Sst2+), or gip cells, but from Sst1+ cells. Furthermore, Mi et al. demonstrated that Sst1+ cells were derived from Krt4+ ductal cells, distinct from notch-responsive ductal cells [113]. A series of bioinformatic analyses of single-cell sequencing data revealed the trajectory of Krt4+ to Sst1+ differentiation [113]. These studies vary from mouse models that illustrate a 70–80% loss of beta cells results in regeneration via the proliferation of surviving cells, while a near complete ablation of beta cells in mice results in alpha–beta cell conversion [93]. Humans do not have this incredible capacity for regeneration upon beta cell injury and, therefore, require insulin therapy upon diabetic disease states and major pancreatic damage. A greater understanding of the pathways for regeneration in zebrafish beta cells and other animal models may lead to novel pathways capable of eliciting beta cell-specific proliferation in humans for improved treatment strategies and better glycemic control.

## 5. Islet Inflammation

Beta cell stress and an obesogenic milieu can also perpetuate systemic and islet inflammation. Mounting evidence now shows that inflammatory pathways become chronically activated in T2D [115]. A landmark study insinuating that inflammation was correlated with diabetes was pioneered by Hotamisiligil and colleagues in the 1990s [116]. TNFα was systemically elevated in four different diabetic mouse models, and the neutralization of TNFα increased insulin sensitivity in peripheral tissues significantly [116]. Since this initial study, other research groups have recapitulated this research, finding elevations in inflammatory cytokines in various diabetic models [117,118].

Furthermore, islet inflammation specifically is a hallmark of T2D. The first evidence of increased immune cell infiltration was found in the db/db mouse and Goto-Kakizaki (GK) rat models for diabetes [119,120,121]. In the db/db mouse, Ehses et al. found using immunostaining that macrophage infiltration was increased in the islet of diabetic db/db mice [119]. Additionally, infiltrating macrophages in the islet of db/db mice were proinflammatory in nature, expressing traditional M1-like polarization markers, and were not positive for M2-like, anti-inflammatory markers, including CD206 and CD301 [122]. However, no differences in neutrophil infiltration in the islet were found [119]. In contrast to heterozygous db/+ mice islets, db/db islets expressed several folds higher levels of cytokines and chemokines, including TNFα, IL1β, CCL2, and CXCL1 [122].

Since then, various animal models and human studies have found an increased number of the inflammatory macrophage markers CD68+ and iNOS+ cells in and around the islets of T2D individuals compared to patients without T2D [119,120]. Conversely, these T2D patients are not positive for markers associated with tissue repair macrophages, such as CD163 and CD204, suggesting an inflammatory interaction [119]. Pathogenic interactions between inflammatory macrophages and beta cells occur in diabetic models, including the secretion of IL1β. Increased islet macrophage IL1β secretion reduces beta cell function and disrupts glycemic control, while treatments reducing IL1β and interfering with the IL-1 pathway restore glycemic control and beta cell function [122].

In ZDF rats, there are elevated CRP and TNFα levels in hyperglycemic rats, signaling increased systemic inflammation [119]. Islet inflammation is also elevated. Jourdan et al. predicted that beta cell failure in the ZDF model is associated with M1 polarized macrophages infiltrating the islets and Nlrp3-ASC inflammasomes in macrophages becoming activated during this infiltration [119]. Islet inflammation and Nlrp3-ASC activation are associated with decreases in insulin secretion. The depletion of macrophages reduces this phenotype and restores insulin secretion and normal glycemic levels [119].

Diet-induced diabetes by the use of a high-energy diet in sand rats significantly elevates the level of thioredoxin-interacting protein (TXNIP) [123]. TXNIP is known to inhibit major antioxidant systems in beta cells and lead to increases in oxidative stress. TXNIP knockdown reduces beta cell intrinsic stress and beta cell loss. IL1β expression is increased in sand rat islets due, at least partly, to hyperglycemia as treatment with phlorizin, a drug reducing blood glucose levels by inhibiting glucose absorption in the gut, reduces IL1β expression in the islets [123,124].

Though macrophages were the predominant immune cells implicated in islet inflammation for many years, neutrophils’ role in the perpetuation of islet inflammation has been called into question. Neutrophils move much faster than macrophages and, during inflammation, may not continue to reside in the area for a long time; hence, a possible reason why other studies have failed to find increases in neutrophil infiltration. Therefore, neutrophils’ role in pathogenesis may be harder to assess. In our diabetic zebrafish model, both macrophages and neutrophils were found to interact with beta cells, leading to a loss of beta cell mass [125]. We discovered the following progression of inflammatory events: (1) the activation of Ripk3 initiated by ER stress in beta cells; (2) Ripk3-dependent induction of *il1b* and other cytokines in beta cells; (3) Il1b-dependent recruitment of Tnfa-secreting macrophages into the islet and Tnfa secretion; (4) Tnfa-dependent induction of *cxcl8a* in beta cells; (5) *cxcl8a*-dependent recruitment of neutrophils into the islet; and (6) loss of beta cells. These key chain events are essential for beta cell loss in our overnutrition model as disruption of any prevents the loss of beta cell number and function. Intriguingly, crosstalk between beta cells, macrophages, and neutrophils is indispensable for beta cell loss. Endogenous stress in beta cells, along with increases in islet inflammation and inflammatory immune cell infiltration, are drivers of beta cell dysfunction and loss [117,125].

These diabetic models show the importance of islet inflammation in beta cell dysfunction and T2D progression. However, islet inflammation is also an essential component in maintaining islet health. For example, lean *il1b*-deficient mice present with glucose intolerance and decreases in insulin expression [126]. Additionally, macrophages have been found to be important in beta cell mass expansion during embryonic development and adulthood [122]. Therefore, it is crucial to view inflammation as a spectrum, wherein it can promote islet health under a range of circumstances. Yet, if the scale tips in excess or chronically, inflammation can significantly undermine beta cell function.

## 6. Decompensation: Beta Cell Death and Loss of Identity

When beta cells are pushed to a certain point of strain, many groups have found that there is a loss of beta cell mass. Loss of beta cell mass can happen through a variety of different mechanisms. The major avenues of beta cell loss are through increased beta cell death and loss of beta cell identity. Loss of beta cell identity occurs when beta cells stop expressing beta cell markers. The beta cell markers include insulin and several transcription factors, including NKX6.1, MAFA, and PDX1 in mice [127]. In T2D human cadavers, a reduction in these transcription factors, NKX6.1, MAFA, and PDX1, was also observed [128]. These studies show that under some conditions of T2D pathophysiology, beta cell loss of identity may occur. However, there is also evidence for loss of beta cell mass by beta cell death. As cell death occurs relatively quickly and cell corpses are rapidly cleared, it is more difficult to detect cell death. Because T2D is such a heterogeneous disease as previously discussed, it is very likely that either cell death, loss of identity, or a combination of both could occur depending upon the pathological setting in humans.

Our lab has found in the diabetic zebrafish model (zMIR) that beta cell death occurs after 3 days of overfeeding with 5% egg yolk emulsion for 8 h per day. The cell loss occurs on the fourth day in a small window during the night, between 2 a.m. and 4 a.m. [117]. Beta cell loss is not because of dedifferentiation, as all beta cells are marked with a stable fluorescent protein and, therefore, cells will continue to possess the marker even if they stop expressing insulin. Indeed, when the fish are immunostained for insulin, all marked cells still express insulin. Therefore, in our model, it appears that cell death occurs instead of dedifferentiation.

Interestingly, the pathway of cell death does not appear to be apoptosis. Apoptotic inhibitors did not stop cell death in our overnutrition-induced diabetes-prone zebrafish, and our lab has not been able to find apoptotic bodies [117]. Canonical necroptosis is not likely to be involved in beta cell death in our model as the effector, MLKL, has not been found in the zebrafish genome [129]. Therefore, beta cell death in our model likely occurs through an alternative pathway that is not canonical apoptosis or necroptosis. Cells may instead utilize one of the many other pathways of necrotic cell death [130,131]. Studies on this topic are currently ongoing and have the potential to provide insights into novel disease processes in diabetes research.

Beta cell necrosis has also been reported in *Psammomys obesus*, desert sand rats, a non-insulin dependent T2D model, which has been described earlier in this review. This model experiences reductions in beta cell mass after three weeks on a high-energy diet [70]. Reductions in beta cell volume were attributed to necrosis as cell membrane rupture, and swollen mitochondria with dilated cisternae of the Golgi complex and the rough ER in the cytoplasm of beta cells were observed, while apoptotic bodies were not found [132,133]. Therefore, necrotic cell death may be a physiologically relevant avenue of beta cell death in a subset of T2D patients. Understanding the pathway of beta cell death in these various animal models may introduce novel drug targets and alternative pathological pathways for disease progression in T2D.

The db/db mouse model experiences a loss of beta cell mass. In a study by Dalbøge and colleagues, beta cell mass declines in db/db mice from 12 weeks of age, with a peak mean value of 4.84 mg average mass, to 34 weeks with 3.3 mg average mass [73]. The number of islets was found to be similar throughout ages 5–24 weeks, with variations being constrained to islet size and not number. Beta cell proliferation was reduced in 24-week-old mice compared to 10-week-old mice via Ki-67 analysis [73]. This study did not find any significant differences in apoptosis, as measured by the use of caspase 3 immunoreactive assays. An alternative study by Puff and colleagues found apoptosis to be increased in db/db mice. However, their studies were performed on mice at earlier time points of 5–12 weeks of age, and they were unable to ensure that these cells were truly beta cells via staining [134] ZDF rats also experience a robust decompensation of beta cell mass, losing more than 50% in some cases [135]. Beta cell loss was thought to be an outcome of increased apoptosis, as increased DNA fragmentation was found in several studies [95,135].

## 7. Future Outlook

T2D is a serious and worsening epidemic affecting the life and health span of humans all around the world. This review characterizes major breakthroughs in disease heterogeneity, beta cell compensation and decompensation, and the role islet inflammation has on pathogenesis in T2D using initial discoveries in rodents and zebrafish models, many of which have paved the way for better treatments for human patients. Although a cannon of research exists in the field, there is still much left to be revealed. For instance, how beta cell loss occurs in humans is unknown. Despite several groups, including Butler et al., have noted a decrease in beta cell mass in cadavers, what cell death pathway(s) are activated in beta cells in vivo have not fully been addressed [90]. Additionally, in humans, it is unknown to what extent beta cell death versus other processes shown in Figure 2 may play a role in the overall decrease in beta cell mass. Zebrafish may pose an advantageous model to address this question, as disease processes can be observed in real time with live imaging.

Many heterogeneous disease pathways can culminate in the outcome of T2D. Therefore, an even greater need to understand a variety of disease processes in patients exists. Pharmacologic, dietary, and surgical treatments for T2D need to be optimized to fit the bio-individuality of a particular patient to have the greatest success and outcomes for the patient. Various animal models need to be utilized, including rodents, zebrafish, and non-human primates, to understand variances in disease development and discover novel screening tests to discern the major pathological pathways driving disease in pre-diabetic and diabetic patients. Combined with the development of pathway-specific treatments, the identification and understanding of these pathways will lead to personalized care and novel treatments for T2D.

## Figures and Tables

**Figure 1 biomedicines-12-00473-f001:**
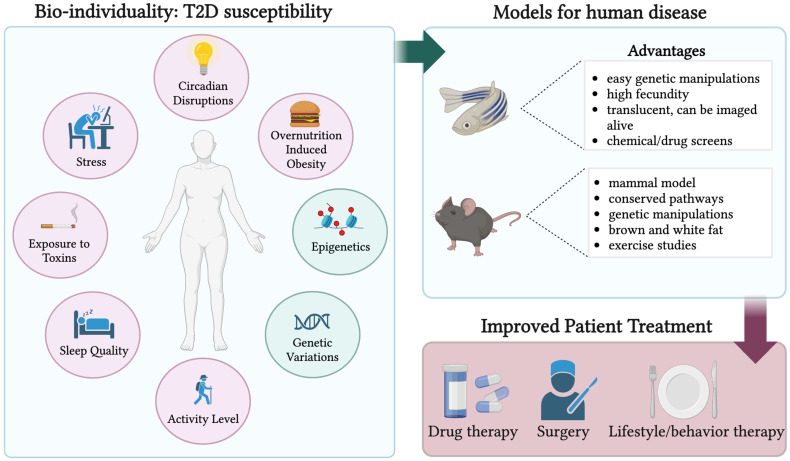
A schematic representing the genetic and environmental factors that play a role in T2D susceptibility. The pink circles represent environmental factors while the blue circles show genetic and epigenetic variations. The right top panel depicts two of many diabetic animal models that can be used to better understand the individual factors encompassed in the circles on the left panel. The advantages of these animal models are listed. The bottom right panel illustrates a transition from animal models to better drug, surgical, and lifestyle/behavioral therapies. Created with BioRender.com (2 February 2024).

**Figure 2 biomedicines-12-00473-f002:**
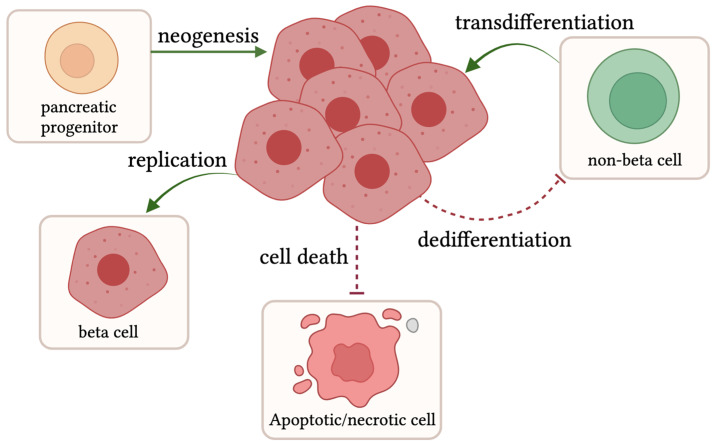
Routes promoting beta cell expansion: Beta cell numbers can expand through different pathways, including neogenesis from pancreatic progenitor cells, intrinsic cell replication, or transdifferentiation of non-beta cells into beta cells, as shown by the green arrows. Other factors promoting beta cell mass expansion are the inhibition of both beta cell death and the dedifferentiation of beta cells. Created with Biorender.com (2 February 2024).

**Table 1 biomedicines-12-00473-t001:** Represents the T2D animal models: zebrafish muscle insulin resistance (zMIR) [64], sand rats [65], Zucker Diabetic Fatty (ZDF) Rats [66], and db/db mice [67], along with their respective genetic alteration or treatment, which elicits the phenotype in the neighboring column. Animal model pros and cons are listed next, along with paper references.

Animal Model	Genetic Alteration/Treatment	Phenotype	Pros/Cons	References
Zebrafish muscle insulin resistance (zMIR) (*Danio rerio*)	Dominant-negative IGF1R diet: 5% egg yolk	Muscle insulin resistance, beta cell compensation, and decompensation	Pros: live imaging, high fecundity, insulin resistance, quick disease progression, drug/genetic alterations easier Cons: non-mammal model	Maddison et al. [64]
Sand rat (*Psammomys obesus*)	High-energy diet	Obese, hyperglycemia, insulin resistance, beta cell compensation, and decompensation	Pros: mammal, insulin resistance, quick disease progression, dyslipidemia Cons: seasonal breeder, lower fecundity, genetic manipulations difficult	Schmidt-Nielsen et al. [65]
Zucker Diabetic Fatty Rat (ZDF) (*Rattus norvegicus*)	Leptin receptor mutation Gln269Pro	Obese, hyperglycemia, beta cell compensation, and decompensation	Pros: mammal, hyperphagic, obese, hyperglycemia, islet structure more comparable to humans Cons: inbred, expensive, longer time to disease	Peterson et al. [66]
db/db mouse (*Mus musculus*)	Leptin receptor mutation 106 nt insertion	Obese, hyperglycemia, beta cell compensation, and decompensation	Pros: mammal, hyperphagic, obese, hyperglycemia Cons: inbred, longer time to disease	Hummel et al. [67]

## Data Availability

Not applicable.

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
