# Peer review of "Animal Models for Understanding the Mechanisms of Beta Cell Death during Type 2 Diabetes Pathogenesis"

_biomedicines, 2024, doi:10.3390/biomedicines12030473_

Round 1
Reviewer 1 Report
Comments and Suggestions for Authors
Well written review.
But it is advised to discuss a short summary about insulin resistance and GLP receptor and the role of brain (hypothalamus) in the pathobiology of T2DM and how it impacts clinical management. Beta cell apoptosis is not always present in 2DM but may occur after long duration of DM. Hence, concentrating on insulin resistance is important.
Comments on the Quality of English Languageok
Author Response
Thank you for your insightful suggestions of including a short summary about insulin resistance and GLP-1 in clinical use in the context of diabetes pathology. We have included a few paragraphs covering these two topics.
a) On insulin resistance, we have included more information in lines 57-66.
b) We have added a section on current drug therapies for T2D treatment, including GLP-1RA and their mechanism of action in lines 88-124.
Reviewer 2 Report
Comments and Suggestions for Authors
In the article titled "Animal models for understanding the mechanisms of beta cell death during type 2 diabetes pathogenesis" the authors highlight the need for diverse T2D animal models to better understand the varied pathologies and treatment responses observed in T2D patients.
The article demonstrates a somewhat superficial approach in its composition. Despite the multitude of available animal models for diabetes, the authors have opted to selectively highlight only a few in this review article. The descriptions provided for each model are notably concise, lacking in depth. Furthermore, the article solely focuses on enumerating the advantages of employing each model, neglecting to address their inherent limitations.
It is crucial for researchers to possess a comprehensive understanding of both the strengths and weaknesses associated with each animal model when selecting the most suitable model for studying a particular disease. Highlighting solely the positive aspects creates an incomplete perspective, potentially leading to an oversight of critical nuances in the interpretation of experimental outcomes. In future reviews, a more balanced presentation, encompassing both the merits and limitations of each model, would contribute to a more nuanced and informative resource for the scientific community.
In Table 1, the authors briefly outline the drawbacks of each model, albeit succinctly. It is important to note that the sole disadvantage attributed to zebrafish is not merely its non-mammalian nature.
In humans, Type 2 diabetes is a polygenic disease wherein environmental factors also contribute to its development. Therefore, it would be beneficial to incorporate one of these mouse models, specifically diet-induced Type 2 diabetes, into the review.
The title of Chapter 4 is "Animal Models for Beta Cell Death," yet the actual content of this chapter is virtually non-existent. Only Table 1 is presented, providing a summary of the preceding chapters' content.
Chapter 5 explores beta cell proliferation, transgenesis, and neogenesis. This prompts the question of their relationship with insulin secretion. Are these newly formed cells functionally active, contributing to increased insulin secretion? The crucial connection lies not only in expanding beta cell mass but in ensuring the adequacy of insulin secretion for maintaining normal glucose homeostasis. It becomes essential to explore how glucose sensitivity is altered in the context of these cellular processes.
Below I have listed minor comments for the article.
Line 34: The final punctuation mark is missing.
Line 43: Use a small initial in the word diabetes.
Line 209: What does D stands for?
Author Response
Thank you for your comments on the review. We have modified the review to reflect your suggestions in the following way:
- We have included an additional section which states the limitations of rodent and zebrafish models. This text can be found in lines 246-262.
- The sand rat is a model which does not have any genetic mutations. This animal develops obesity and metabolic disease due to diet. However, this may not have been clear. To help clarify the text, we have stated this more clearly in the review by adding the lines 209-211.
- Chapter 4 was meant to be a figure title but to remove confusion, we have deleted the figure title.
- We have expanded on how increased beta cell mass also increases insulin content and indicated how increased insulin secretion in the face of hyperglycemia can increase beta cell mass. This is shown in lines 353-357. We have also expanded upon the compensation section by adding lines 278-284 and an additional figure.
- We have indicated where the reader can find reviews that cover a multitude of different animal models. This text can be found in lines 185-187.
- The final punctuation mark is added in line 35.
- Diabetes has been lower cased in line 43.
- Line 209, we are unsure what D you are referring to. Maybe it is not the correct line number?
Reviewer 3 Report
Comments and Suggestions for Authors
To state the gap of research / review considering two similar reviews have addressed the topic in 2016 and 2021.
Kottaisamy, C. P. D., Raj, D. S., Prasanth Kumar, V., & Sankaran, U. (2021). Experimental animal models for diabetes and its related complications-a review. Laboratory animal research, 37(1), 23. https://doi.org/10.1186/s42826-021-00101-4
Pandey, S., Chmelir, T., & Chottova Dvorakova, M. (2023). Animal Models in Diabetic Research-History, Presence, and Future Perspectives. Biomedicines, 11(10), 2852. https://doi.org/10.3390/biomedicines11102852
Line 66: …there a multitude of factors… - there is a
Section 4: Missing explanation. Only one Table is provided.
Section 5: Beta cell compensation for insulin resistance with subsections 5.1, 5.2, 5.3
e.g 5.1 Mechanisms for expansion of β cell mass and 5.2 Enhanced β cell performance
A schematic diagram(s) of mechanisms of β cell compensation for insulin resistance would help our understanding of compensatory mechanisms
To include a section of Challenges and future perspectives
Comments on the Quality of English Language-
Author Response
Thank you for your insightful suggestions and comments. To address these, we have:
a) included where the reader can find reviews that cover a multitude of different animal models, lines 185-187.
b) addressed the gap in knowledge in lines 513-518.
c) addressed the grammatical error in line 74.
d) removed the title header for the table in section 4 to omit confusion.
e) separated section 5 into subsections including rodent, human and zebrafish beta cell mass expansion.
f) included an additional diagram of beta cell compensation and included more about the pathways of beta cell mass expansion as seen in lines 386-391.
g) included a section stating the limitations/challenges with these diabetic models in lines 246-262.
Round 2
Reviewer 1 Report
Comments and Suggestions for Authors
accept
Comments on the Quality of English Languageok
Reviewer 2 Report
Comments and Suggestions for Authors
comments are taken into account